# Effectiveness of Initial Troponin I and Brain Natriuretic Peptide Levels as Biomarkers for Predicting Delayed Neuropsychiatric Sequelae in Patients with CO Poisoning: A Retrospective Multicenter Observational Study

**DOI:** 10.3390/jpm13060921

**Published:** 2023-05-30

**Authors:** Myung Hyun Jung, Juncheol Lee, Jaehoon Oh, Byuk Sung Ko, Tae Ho Lim, Hyunggoo Kang, Yongil Cho, Kyung Hun Yoo, Sang Hwan Lee, Chang Hwan Sohn, Won Young Kim

**Affiliations:** 1Department of Emergency Medicine, Hanyang University Medical Center, Seoul 04763, Republic of Korea; jmhaudgus@gmail.com (M.H.J.); ojjai@hanyang.ac.kr (J.O.); postwinston@gmail.com (B.S.K.); erthim@gmail.com (T.H.L.); emer0905@gmail.com (H.K.); joeguy@hanmail.net (Y.C.); camcalvin@naver.com (K.H.Y.); sangwhan123@hanyang.ac.kr (S.H.L.); 2Department of Emergency Medicine, College of Medicine, Hanyang University, Seoul 04763, Republic of Korea; 3Department of Emergency Medicine, University of Ulsan College of Medicine Asan Medical Center, Seoul 05505, Republic of Korea; schwan97@gmail.com (C.H.S.); wonpia73@naver.com (W.Y.K.)

**Keywords:** carbon monoxide poisoning, delayed neuropsychiatric sequelae, troponin, brain natriuretic peptide

## Abstract

Background: Delayed neuropsychiatric sequelae (DNS) are a severe complication of carbon monoxide (CO) poisoning, and predicting DNS is difficult. This study aimed to investigate whether cardiac markers can be used as biomarkers to predict DNS occurrence following acute CO poisoning. Methods: This was a retrospective observational study that included patients with acute CO poisoning who visited two emergency medical centers in Korea from January 2008 to December 2020. The primary outcome was whether the occurrence of DNS was associated with laboratory results. Results: Of the 1327 patients with CO poisoning, 967 patients were included. Troponin I and BNP were significantly higher in the DNS group. As a result of multivariate logistic regression analysis, it was found that troponin I, mentality, creatine kinase, brain natriuretic peptide, and lactate levels independently influenced DNS occurrence in CO poisoning patients. The adjusted odds ratios for DNS occurrence were 2.12 (95% CI 1.31–3.47, *p* = 0.002) for troponin I and 2.80 (95% CI 1.81–3.47, *p* < 0.001) for BNP. Conclusion: Troponin I and BNP might be useful biomarkers for predicting the occurrence of DNS in patients with acute CO poisoning. This finding can help to identify high-risk patients who require close monitoring and early intervention to prevent DNS.

## 1. Introduction

Acute carbon monoxide (CO) poisoning is one of the most common types of fatal poisoning worldwide, and its incidence has remained constant over the past 25 years [1]. CO, a colorless and odorless gas, is rapidly absorbed into the blood through the lungs and causes nerve and cardiac damage through hypoxia and inflammatory mechanisms. CO has a more than 200-fold greater affinity for hemoglobin than oxygen, resulting in a leftward shift in the oxyhemoglobin dissociation curve. Binding of CO to heme proteins damages mitochondrial function and causes tissue hypoxia, which leads to oxidative stress, cell necrosis, and apoptosis. Furthermore, the free carbon monoxide in plasma binds to platelet heme proteins, causing platelet-neutrophil aggregation and neutrophil degranulation. This produces reactive oxygen species through the production of xanthine oxidase, causing lipid peroxidation, apoptosis, and ongoing inflammatory reactions. The neurological and myocardial damage associated with this pathological mechanism is the major cause of mortality and morbidity [2,3,4].

Delayed neuropsychiatric sequelae. (DNS) is a neurological complication that occurs 2–40 days after complete recovery from acute CO poisoning [4]. DNS symptoms range from mild to severe, including headache, seizures, altered consciousness, concentration problems, cognitive impairment, emotional irresponsibility, personality changes, amnesia syndrome, dementia, psychosis, gait disturbances, movement disorders, chorea, apraxia, agnosia, asthenia, peripheral neuropathy, urinary incontinence, and even a vegetative state [5,6]. The absence of universal diagnostic criteria makes it difficult to precisely determine the incidence of DNS, with reported incidence rates varying from 7% to 40% across different studies [3,7]. The lack of diagnostic criteria leads to delays in diagnosis, which in turn can lead to delays in treatment and can negatively impact treatment outcomes. Therefore, it is important to identify patients who may develop DNS early on and establish appropriate treatment strategies. Furthermore, as the mortality rate of CO poisoning is decreasing, preventing DNS is considered to be the primary goal of CO poisoning treatment [1,3,5,7].

Previous studies have suggested low initial GCS score, longer exposure to CO and abnormal magnetic resonance imaging (MRI) findings as independent factors for predicting the occurrence of DNS [3,5,8]. However, due to various factors such as the lack of MRI equipment, the patient’s hemodynamic instability, including mentation, and the patient’s refusal to have an MRI, it is difficult to perform MRI in all patients with CO poisoning. If the time between the onset of CO poisoning and MRI is too short, characteristic findings of MRI due to CO poisoning may not be detected. Therefore, it is necessary to identify effective biomarkers that are readily obtainable in most patients for predicting DNS. Currently, studies on serum lactate, anion gap, cardiac enzymes, creatine kinase, neuron specific enolase (NSE), etc., are being conducted, but the size of the studies is relatively small, and consistent results have not been obtained [3,8,9,10]. As described above, the main target of the damage mechanism of CO poisoning is neurological and cardiac damage, so it can be assumed that there is a correlation between the extent of cardiac damage and neurological damage. One of the biomarkers that reflect cardiac injury due to CO poisoning is brain natriuretic peptide (BNP). BNP is released from cardiac myocytes in response to ventricular wall stress and stretch, indicating myocardial injury and dysfunction. Studies have reported that BNP levels are elevated in patients with acute CO poisoning and are associated with the severity of cardiac injury [11]. In addition, troponin is known as another effective biomarker for detecting cardiac injury and has the advantage of being easily identified in all patients [12]. Therefore, this study aims to investigate the usefulness of troponin I and BNP for predicting the occurrence of DNS by countering the limitations of previous studies.

## 2. Materials and Methods

### 2.1. Study Design and Setting

This study was conducted as a multicenter retrospective observational study and included the reviewing of electronic medical records of patients with acute CO poisoning who visited two tertiary-care emergency medical centers from January 2008 to December 2020. This study was ethically approved by the Institutional Review Board of the Hanyang University Hospital and Asan Medical Center in March 2023 (Seoul, Republic of Korea, HYUH 2023-02-023-001).

### 2.2. Study Participants

Participants in this study were selected based on specific criteria. The following patients were included in this study: (1) patients who had an adequate history of CO poisoning, and (2) the initial carboxyhemoglobin measurement was over 5% in nonsmokers and over 10% in smokers when visiting the emergency room. The exclusion criteria were as follows: (1) patients aged <18 years, (2) underlying abnormal neurologic deficits before poisoning, (3) patients who did not undergo troponin I analysis or other laboratory tests, (4) patients who had gone into cardiac arrest as a result of CO poisoning, and (5) patients for whom it could not be determined whether DNS had occurred.

### 2.3. Data Collection

Data for the study were obtained from electronic medical records and a picture archiving and communication system (PACS, Centricity, GE Healthcare, Milwaukee, WI, USA). A list of patients who visited the emergency room and were assigned the T58 code (International Classification of Diseases, ICD-10; toxic effect of CO) during the study period was used for patient selection. The variables extracted through the electronic medical records and PACS were as follows: (1) baseline characteristics: age, sex, vital signs, and comorbidities; (2) laboratory results from the initial blood samples of patients who visited the emergency center and discharge GCS, coingestion of drugs or alcohol, intentionality of exposure, duration of exposure, the number of hyperbaric oxygen therapy (HBOT) sessions, presence of DNS, and abnormal findings on brain MRI.

### 2.4. Primary Outcome

In this study, the primary outcome was whether the occurrence of DNS was associated with laboratory results. We also investigated factors that affected DNS. However, there are no generally valid diagnostic criteria at this time [13,14]. In this study, we detected patients with disorientation or cognitive impairment within six weeks of acute exposure who had a high intensity of brain white matter on brain MRI, which is typical of DNS [15,16]. Disorientation and cognitive impairment were measured using the Mini-Mental State Examination (MMSE) [17]. The MMSE was administered by trained emergency medicine physicians.

### 2.5. Statistics

Statistical analysis was conducted using Excel 2016 for data compilation and R (version 3.6.1) for analysis. The normality of the datasets was assessed using Kolmogorov-Smirnov tests. Descriptive statistics were used to summarize the baseline characteristics of the study participants. Categorical variables were presented as frequencies and percentages. Normally distributed data are presented as the mean ± standard deviation (SD), whereas non-normally distributed data are presented as medians with interquartile ranges (IQR). Independent *t*-tests or Mann–Whitney U tests were used for continuous variable comparisons, and chi-square or Fisher’s exact tests were employed for categorical variables. To minimize the effect of confounding bias given our study design, propensity score matching methodology was employed. Statistical significance was considered at a *p*-value less than 0.05.

To assess the prognostic performance of laboratory results in predicting DNS occurrence, the area under the receiver operating characteristic curve (AU-ROC) was analyzed. Cutoff values were determined using Youden’s index, which aimed to maximize specificity in predicting DNS based on the ROC curve analysis.

Multivariable logistic regression analysis was performed to identify the association between laboratory results and DNS occurrence. The results of logistic regression analysis were presented as odds ratios (ORs) with corresponding 95% confidence intervals (CIs). Continuous variables, representing laboratory tests included in the multivariable logistic regression, were converted into categorical variables based on the cutoff value derived from Youden’s index on the ROC curve. Age was also transformed into a categorical variable using a cutoff value of 65 years.

## 3. Results

From January 2008 to December 2020, 1327 patients were enrolled in the registry. We excluded patients under the age of 18 (n = 25), those lost to follow-up (n = 101), and those with missing data for certain variables (n = 234); comorbidity (n = 11), loss of consciousness (n = 18), Seizure (n = 25), laboratory findings (n = 180). A total of 967 patients were eligible for this study (Figure 1).

The baseline characteristics of the group of patients with and without DNS are summarized in Table 1. A total of 597 (61.7%) were male, and the median age was 41.0 years (IQR, 31.0–54.0). Of the 967 patients who met the inclusion criteria, 171 patients experienced DNS. 

When comparing the age distribution of individuals in the non-DNS and DNS groups based on their baseline characteristics, statistical analysis revealed a *p*-value of less than 0.001. This significant finding implies that the disparity observed in biomarkers between the two groups might be influenced by the variation in age. To account for this potential confounding factor, an additional analysis was performed to match the ages of individuals in both groups, as presented in Table 2.

The predictive performance of troponin I and BNP for the occurrence of DNS in CO poisoning patients was analyzed using the ROC curve (Figure 2 and Figure 3). The AUC of troponin I was 0.718 (95% CI: 0.670–0.766). The optimal cutoff value of the troponin I level for predicting DNS occurrence using Youden’s index was calculated as 0.056 ng/mL. In addition, the AUC of troponin I was 0.708 (95% CI: 0.660–0.756). The optimal cutoff value of the BNP level for predicting DNS occurrence using Youden’s index was calculated as 34.5 pg/mL.

The following variables were included in a multivariate logistic regression analysis of the data of CO poisoning patients with DNS (Table 3): age, loss of consciousness, mentality, seizure, HBOT, WBC, troponin I, CK, BNP, and lactate levels. The factors that independently influenced DNS occurrence in CO poisoning patients were mentality and troponin I, CK, BNP, and lactate levels. The adjusted odds ratio for DNS occurrence and troponin I was 2.14 (95% CI, 1.31–3.49, *p* = 0.002). In addition, the adjusted odds ratio for DNS occurrence and BNP was 2.31 (95% CI, 1.49–3.57, *p* < 0.001).

## 4. Discussion

This study is a retrospective analysis of acute carbon monoxide poisoning patients who visited two tertiary university hospitals between January 2008 and December 2020. It included 967 carbon monoxide poisoning patients, of whom 171 (17.7%) experienced delayed neurological sequelae (DNS). The mortality rate due to CO poisoning was only 0.4%. We performed multivariable logistic regression analysis to identify the association between laboratory results and DNS occurrence. Troponin I and BNP were significantly higher in the DNS group and were identified as independent biomarkers for predicting DNS occurrence. In addition, the AUC of troponin I was 0.718 (95% CI: 0.670–0.766), and the AUC of BNP was 0.708 (95% CI: 0.660–0.756). The optimal cutoff value of the troponin I and BNP were 0.056 ng/mL and 34.5pg/mL, respectively.

CO poisoning is one of the most common inhalation poisonings, and its global incidence has remained steady over the past 25 years, although there are significant regional differences [1]. In Korea, prior to 2006, unintentional poisoning due to heating equipment such as coal boilers accounted for most cases. However, since the 2008 suicide case involving a famous person and carbon monoxide poisoning, the number of patients with carbon monoxide poisoning due to suicide attempts has increased dramatically. From 2006 to 2015, the annual incidence rate of carbon monoxide poisoning in Korea more than doubled from 4.4 cases per 100,000 to 10.4 cases per 100,000 [18]. Between 2006 and 2012, the number of patients with intentional carbon monoxide poisoning increased by 3183% [19]. Lee et al. reported patients with intentional carbon monoxide poisoning have significantly higher rates of moderate severity, intensive care unit admission, and emergency room mortality compared to those with unintentional carbon monoxide poisoning [20]. Therefore, it can be said that the incidence and severity of carbon monoxide poisoning patients in Korea are both increasing. However, despite 50% of the total population being concentrated in the metropolitan area, only six HBOT centers operate in the metropolitan area, mostly using single-chamber HBOT equipment. This leads to delayed HBOT for carbon monoxide poisoning patients, especially those with severe poisoning or those who have decreased consciousness due to alcohol or drug co-ingestion or who are receiving ventilator care. In fact, alcohol co-ingestion was present in 43.3% of patients and drug co-ingestion was present in 23.3% of patients in this study [20]. Therefore, it is important to identify patients who may experience severe complications or delayed neurological sequelae due to carbon monoxide poisoning early and to administer HBOT or transfer them quickly to a facility where it can be performed.

In the treatment of CO poisoning, early prediction of DNS, providing appropriate education, and establishing treatment strategies for patients are highly important. Despite being a potentially life-threatening condition that can cause permanent brain damage, DNS can be confused with other conditions, such as dementia, Parkinson’s disease, and acute ischemic stroke, due to the absence of clear diagnostic criteria and characteristic symptoms, and patients may not be aware of the occurrence of DNS [15]. Although there is currently no definitive treatment for DNS, recent studies have reported that early administration of HBOT can reduce the frequency of DNS and has a beneficial effect on improving symptoms when HBOT is initiated early after DNS occurrence [21,22,23]. Predicting DNS occurrence in advance and initiating treatment promptly when it occurs play a crucial role in improving patient outcomes.

In carbon monoxide poisoning, it is well known that tissue hypoxia causes damage to organs such as the heart and brain, which are vulnerable to ischemic injury, and a series of inflammatory cascades occur via NO and ROS, leading to myocardial and neurological damage independent of hypoxia [2,3,4]. Given these characteristics, attempts have been made to explore the relationship between myocardial damage in carbon monoxide poisoning and neurological outcomes in patients. Koga et al. reported that in a study of 786 carbon monoxide poisoning patients from 1990 to 2011, approximately 42% of patients had myocardial damage, and the presence of myocardial damage was associated with the occurrence of persistent/delayed neurological sequelae [24]. Additionally, recent studies have reported that the presence of myocardial damage is associated with the prognosis of carbon monoxide poisoning, including mortality and neurological outcomes [25,26].

Troponin I is a protein specific to myocardial tissue that regulates the contraction and relaxation of cardiac muscle. Damage or death of cardiac muscle cells results in the leakage of troponin I into the bloodstream, causing an increase in the concentration of serum troponin I, which serves as an indicator of the presence and extent of myocardial damage. Troponin I elevation can be observed not only in cases of carbon monoxide poisoning but also in many critically ill patients, including those with ischemic stroke, intracerebral hemorrhage, sepsis, and recovery of spontaneous circulation (ROSC) after cardiac arrest [27,28]. Many previous studies have consistently shown that troponin I elevation predicts short- and long-term mortality and neurological prognosis in various disease states [28,29,30,31,32,33,34]. Troponin I elevation has been reported to predict short-term mortality in acute coronary syndrome, sepsis, pulmonary embolism, ischemic stroke, and subarachnoid hemorrhage and to predict neurological prognosis in ischemic stroke and subarachnoid hemorrhage [28,30,31,32,33,34]. Yanagiha et al. reported that myocardial damage in moderate to severe carbon monoxide poisoning is a single factor that increases long-term mortality [21], and Kao et al. reported that myocardial damage in severe carbon monoxide poisoning predicts short-term mortality and neurological sequelae at discharge [35]. Therefore, troponin I elevation is considered an important factor in predicting mortality and neurological prognosis in various severe illnesses, but its impact on the development of DNS in general carbon monoxide poisoning has not yet been clearly elucidated.

In this study, BNP was identified as an independent predictor of DNS occurrence in multivariate analysis along with troponin I. BNP and NT-pro BNP are endogenous cardiac hormones that can be secreted during myocardial stress. The usefulness of BNP in predicting myocardial damage in CO poisoning has been reported in previous studies. Davutoglu et al. reported that NT-pro BNP was useful as an early marker of myocardial damage in acute CO poisoning [34]. Yücel et al. compared the usefulness of BNP and troponin I levels in determining the severity of myocardial injury caused by CO poisoning in an animal experiment study. Troponin I showed a significant increase within 6 h after CO poisoning, while BNP showed a statistically significant increase compared to the control group within 1 h, indicating its usefulness as an early biomarker for myocardial damage [11]. Moon et al. investigated the association between troponin I and NT-pro BNP and long-term neurological outcomes in 220 patients with CO poisoning and found that NT-pro BNP was an independent factor in multivariate analysis [36]. Considering that troponin I increases 4–6 h after myocardial damage occurs, these results suggest that BNP may be more useful than troponin I in determining the presence of myocardial damage in CO poisoning and may be a useful biomarker in predicting DNS occurrence.

The levels of serum CK and lactate were significantly higher in patients with DNS. Lactate elevation occurs through various mechanisms. Systemic hypoxia and mitochondrial damage caused by CO poisoning induce anaerobic glycolysis, and other complex factors such as inflammatory mediators, catecholamines, and Na/K adenosine triphosphatase activity stimulation lead to increased serum lactate levels. Additionally, mitochondrial dysfunction and decreased energy production due to hypoxia can increase the concentrations of creatine phosphate and ADP (Adenosine diphosphate) in muscle and activate creatine kinase, which breaks down creatine to produce ATP (Adenosine triphosphate) [2,3,4]. Furthermore, prolonged states of decreased consciousness due to carbon monoxide poisoning or coingested drugs, can cause pressure sores and muscle necrosis, leading to rhabdomyolysis and a significant increase in CK. Serum lactate levels have long been used as an indicator of tissue hypoxia, severity, and prognosis in critically ill patients suffering from conditions such as sepsis, major trauma, cardiogenic shock, post-ROSC, acute respiratory failure, etc., and may serve as an indicator of severe poisoning in CO poisoning [37]. Therefore, the elevation of serum CK and lactate levels may be associated with the severity and duration of CO poisoning and decreased consciousness, which are known factors related to DNS occurrence in previous studies [3,5,8]. Therefore, it can be presumed that the elevation of CK and serum lactate levels may indirectly predict the occurrence of DNS.

Our study has important clinical implications for the treatment of CO poisoning patients. Previous studies on biomarkers for predicting DNS were single-institution studies, whereas our study is a multi-institutional study involving patients from two tertiary hospitals and is, to our knowledge, a relatively large scale study. Furthermore, our study demonstrated the association between myocardial injury and DNS in general CO poisoning patients, which has not been clearly shown before. Several studies have reported that abnormal findings on DW-MRI and prolonged CO exposure, as well as changes in consciousness, were significant factors associated with DNS occurrence [3,5]. However, it is difficult to determine whether changes in consciousness are solely attributable to CO poisoning or are also attributable to co-ingestion of other drugs given the high prevalence of co-ingestion of other drugs in our patient population, and it is often challenging to ascertain the duration of CO exposure due to limitations in obtaining medical history. MRI may also be difficult to perform in all patients for various reasons, and the timing of MRI may vary among hospitals. In fact, in our study, MRI was only performed in 65.7% of patients. Troponin I and BNP, both of which can be easily measured in the early stages for almost all patients, offer clinicians an objective indicator for predicting the occurrence of DNS using the cutoff values proposed in this study. As a result, clinicians can benefit significantly when making crucial treatment decisions to enhance the neurological prognosis of patients. These decisions may involve early implementation of more aggressive therapies, such as HBOT, or extended durations of HBOT administration, or even transferring patients to hospitals where appropriate treatment options are available.

This study had several limitations. First, this study was a multicenter study with relatively large sample sizes, but it is a retrospective observational study using medical records. Second, since this study was conducted in one country, it was limited in terms of race and nationality. However, it was more representative of South Korea. Third, although we used a characteristic finding on brain MRI and the MMSE to objectively diagnose DNS, it may not be appropriate for predicting a wide range of DNS. Finally, of 1327 CO poisoning patients, only 967 patients were included. A total of 234 patients with missing blood test results and 101 patients lost to follow-up were excluded. The exclusion of a large number of patients may have created an opportunity for selection bias.

## 5. Conclusions

In this study, elevated levels of troponin I and BNP were associated with high DNS occurrence. Biomarkers representing cardiac injury, such as troponin I and BNP, were significant independent predictors of DNS. These biomarkers may be useful in predicting the occurrence of DNS.

## Figures and Tables

**Figure 1 jpm-13-00921-f001:**
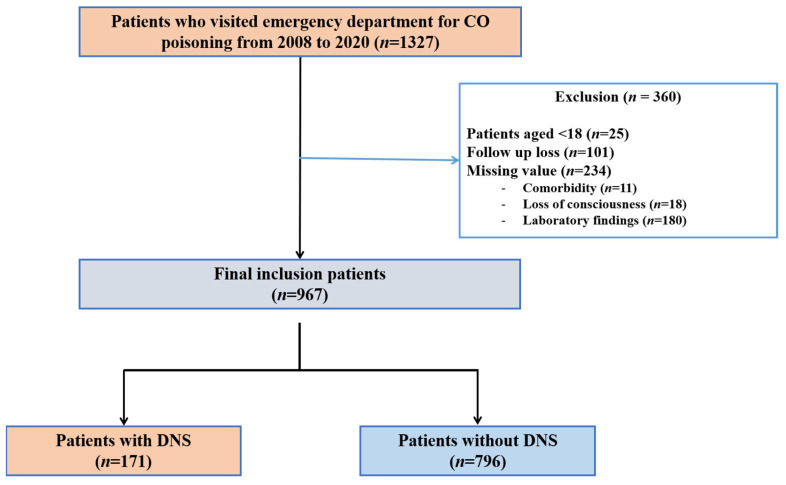
Flow chart of the study. CO, carbon monoxide; DNS, delayed neuropsychiatric sequelae.

**Figure 2 jpm-13-00921-f002:**
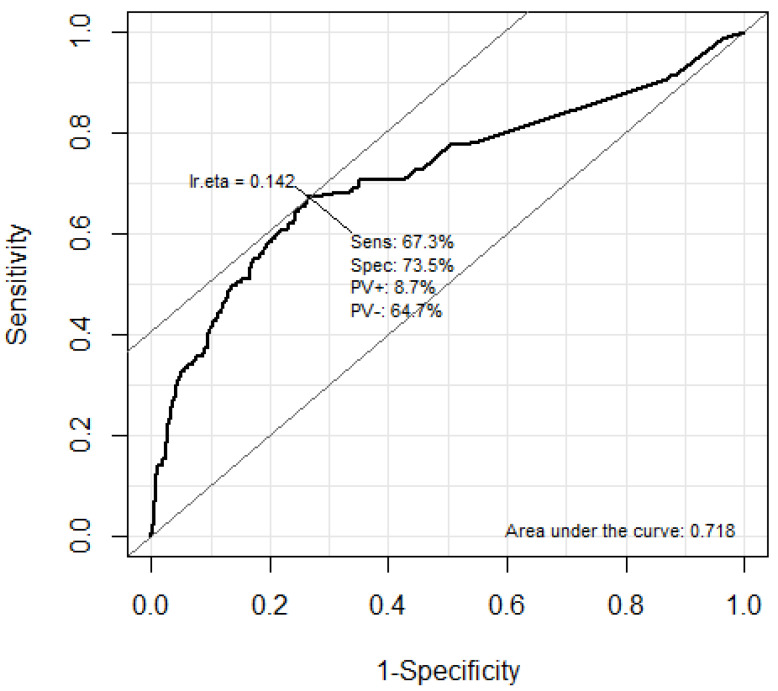
Areas under the ROC curves of Troponin I levels for predicting DNS. The optimal cutoff value of the troponin I level for predicting DNS using the Youden’s index was calculated as 0.056g/mL. ROC, receiver operating characteristic; DNS, delayed neuropsychiatric sequelae.

**Figure 3 jpm-13-00921-f003:**
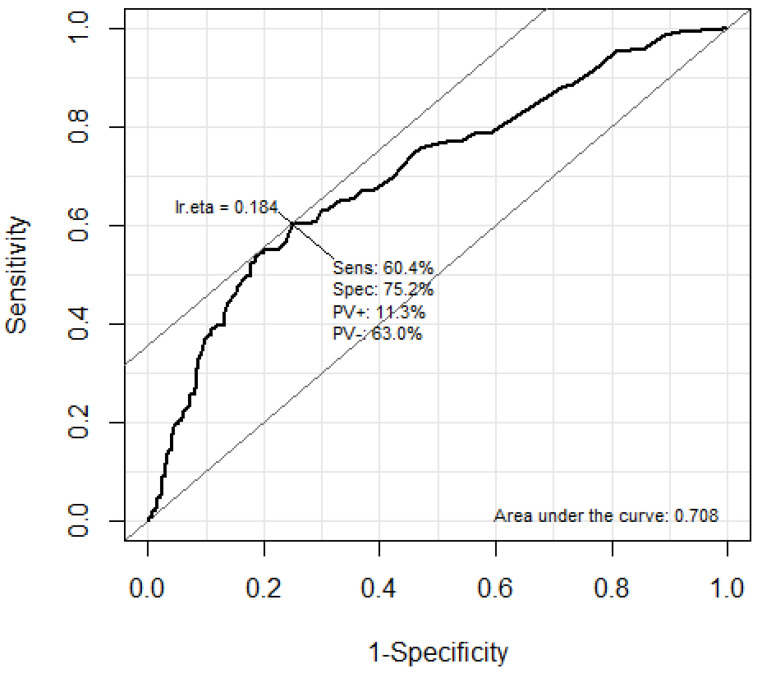
Areas under the ROC curves of BNP levels for predicting DNS. The optimal cutoff value of the BNP level for predicting DNS using the Youden’s index was calculated as 34.5 pg/mL. ROC, receiver operating characteristic; DNS, delayed neuropsychiatric sequelae; BNP, Brain natriuretic peptide.

**Table 1 jpm-13-00921-t001:** Baseline characteristics of the study population.

	Total (n = 967)	Non-DNS (n = 796)	DNS (n = 171)	*p*-Value
Male	597 (61.7%)	478 (60.1%)	119 (69.6%)	0.025
Age (years)	41.0 [31.0; 54.0]	40.0 [31.0; 52.0]	46.0 [35.0; 58.0]	<0.001
Intentionality	665 (68.8%)	548 (69.5%)	117 (68.8%)	0.944
Comorbidity				
HTN	141 (14.6%)	97 (12.2%)	44 (25.7%)	<0.001
DM	85 (8.8%)	65 (8.2%)	20 (11.7%)	0.183
CKD	8 (0.8%)	8 (1.0%)	0 (0.0%)	0.395
CAD	13 (1.3%)	7 (0.9%)	6 (3.5%)	0.019
Current smoker	147 (15.2%)	110 (13.8%)	37 (21.6%)	0.014
Mentality on ED arrival				<0.001
Alert	524 (54.2%)	463 (58.2%)	61 (35.7%)	
Verbal	247 (25.5%)	202 (25.4%)	45 (26.3%)	
Pain	165 (17.1%)	110 (13.8%)	55 (32.2%)	
Unresponsive	31 (3.2%)	21 (2.6%)	10 (5.8%)	
LOC	647 (66.9%)	511 (64.2%)	136 (79.5%)	<0.001
Seizure	25 (2.6%)	21 (2.6%)	4 (2.3%)	1
Systolic blood pressure (mmHg)	125.5 [112.0; 139.0]	126 [112.0; 138.0]	125 [110.0; 145.0]	0.868
Co-ingestion				
Alcohol	419 (43.3%)	345 (43.3%)	74 (43.3%)	1
Drugs	225 (23.3%)	187 (23.5%)	38 (22.2%)	0.797
Brain injury on MRI	163 (25.7%)	57 (11.8%)	106 (70.7%)	<0.001
HBO treatment, done	810 (83.8%)	657 (82.5%)	153 (89.5%)	0.034
HBO treatment number	1.0 [1.0; 3.0]	1.0 [1.0; 3.0]	2.0 [1.0; 3.0]	<0.001
Laboratory finding				
pH	7.4 [7.4; 7.4]	7.4 [7.4; 7.4]	7.4 [7.4; 7.4]	0.489
Base excess (mmol/L)	0.0 [−3.0; 1.6]	0.1 [−2.5; 1.9]	−1.4 [−4.7; 1.1]	<0.001
Lactate (mmol/L)	1.9 [1.0; 3.6]	1.9 [1.0; 3.4]	2.4 [1.4; 4.1]	<0.001
Bicarbonate (mmol/L)	24.0 [21.0; 26.0]	24.0 [21.2; 26.0]	22.0 [19.0; 25.0]	<0.001
COHb (%)	21.3 [8.6; 35.0]	21.5 [9.0; 35.3]	20.7 [7.7; 31.5]	0.424
WBC (×10^3^/mm^3^)	10.5 [7.9; 14.8]	10.0 [7.6; 13.9]	13.2 [9.3; 17.3]	<0.001
Hb (g/dL)	14.7 [13.2; 15.9]	14.6 [13.2; 15.8]	15.0 [13.7; 16.2]	0.004
Platelet (×10^3^/mm^3^)	239.0 [206.0; 282.0]	239.0 [206.0; 280.0]	237.0 [205.0; 290.0]	0.978
CRP (mg/dL)	0.2 [0.1; 0.4]	0.2 [0.1; 0.3]	0.4 [0.2; 2.2]	<0.001
BUN (mg/dL)	13.0 [10.1; 17.0]	13.0 [10.0; 16.0]	16.2 [12.4; 22.0]	<0.001
Cr (mg/dL)	0.8 [0.6; 1.0]	0.8 [0.6; 0.9]	0.9 [ 0.8; 1.2]	<0.001
CK (U/L)	127.0 [79.0; 306.0]	119.0 [75.0; 216.0]	424.5 [120.0; 2833.0]	<0.001
Troponin-I (ng/mL)	0.0 [0.0; 0.2]	0.0 [0.0; 0.1]	0.3 [0.0; 1.9]	<0.001
BNP (pg/mL)	17.0 [9.0; 51.0]	14.0 [8.0; 34.0]	53.0 [16.0; 137.0]	<0.001
Serum osmolality (mOsm/kg)	296.0 [291.0; 309.0]	296.0 [290.0; 309.0]	298.0 [293.0; 308.0]	0.157
Mortality	4 (0.4%)	2 (0.3%)	2 (1.2%)	0.005

Categorical and continuous variables are represented by number (%) and median ± interquartile range, respectively. The two groups were compared using the Mann–Whitney U test for continuous variables, and the chi-square test for categorical variables. HTN, Hypertension; DM, Diabetes mellitus; CKD, Chronic kidney disease; CAD, Coronary artery disease; LOC, loss of consciousness; COHb, Carboxyhemoglobin; WBC, White blood cell; Hb, Hemoglobin; CRP, C-reactive protein; BUN, Blood urea nitrogen; SD, standard deviation; Cr, creatinine; CK, creatinine kinase; BNP, brain natriuretic peptide; *p*-value < 0.05 is significant.

**Table 2 jpm-13-00921-t002:** Age-matched baseline characteristics of the study population.

	Total (n = 342)	Non-DNS (n = 171)	DNS (n = 171)	*p*-Value
Male	220 (64.3%)	101 (59.1%)	119 (69.6%)	0.055
Age (years)	46.0 [35.0; 58.0]	46.0 [35.0; 58.0]	46.0 [35.0; 58.0]	1
Intentionality	235 (69.3%)	118 (69.8%)	117 (68.8%)	0.935
Comorbidity				
HTN	76 (22.2%)	32 (18.7%)	44 (25.7%)	0.153
DM	38 (11.1%)	18 (10.5%)	20 (11.7%)	0.863
CKD	2 (0.6%)	2 (1.2%)	0 (0.0%)	0.478
CAD	8 (2.3%)	2 (1.2%)	6 (3.5%)	0.283
Current smoker	59 (17.3%)	22 (12.9%)	37 (21.6%)	0.045
Mentality on ED arrival				<0.001
Alert	155 (45.3%)	94 (55.0%)	61 (35.7%)	
Verbal	89 (26.0%)	44 (25.7%)	45 (26.3%)	
Pain	85 (24.9%)	30 (17.5%)	55 (32.2%)	
Unresponsive	13 (3.8%)	3 (1.8%)	10 (5.8%)	
LOC	246 (71.9%)	110 (64.3%)	136 (79.5%)	0.003
Seizure	4 (1.2%)	0 (0.0%)	4 (2.3%)	0.131
Systolic blood pressure (mmHg)	127.0 [112.0; 141.0]	127 [114.0; 140.0]	125 [110.0; 145.0]	0.949
Co-ingestion				
Alcohol	144 (42.1%)	70 (40.9%)	74 (43.3%)	0.742
Drugs	77 (22.5%)	39 (22.8%)	38 (22.2%)	1
Brain injury on MRI	122 (48.0%)	16 (15.4%)	106 (70.7%)	<0.001
HBO treatment, done	298 (87.1%)	145 (84.8%)	153 (89.5%)	0.258
HBO treatment number	2.0 [1.0; 3.0]	1.0 [1.0; 2.0]	2.0 [1.0; 3.0]	<0.001
Laboratory finding				
pH	7.4 [7.4; 7.4]	7.4 [7.4; 7.4]	7.4 [7.4; 7.4]	0.922
Base excess (mmol/L)	−0.6 [−3.5; 1.5]	0.0 [−2.3; 2.0]	−1.4 [−4.7; 1.1]	<0.001
Lactate (mmol/L)	2.2 [1.1; 3.7]	2.0 [0.9; 3.4]	2.4 [1.4; 4.1]	0.006
Bicarbonate (mmol/L)	23.6 [20.0; 25.9]	24.2 [22.0; 26.0]	22.0 [19.0; 25.0]	<0.001
COHb (%)	21.1 [8.3; 34.5]	21.8 [9.2; 35.5]	20.7 [7.7; 31.5]	0.397
WBC (×10^3^/mm^3^)	11.2 [8.6; 15.8]	9.9 [8.1; 13.8]	13.2 [9.3; 17.3]	<0.001
Hb (g/dL)	14.8 [13.6; 16.0]	14.5 [13.3; 15.7]	15.0 [13.7; 16.2]	0.015
Platelet (×10^3^/mm^3^)	234.0 [197.0; 280.5]	230.0 [190.0; 275.0]	237.0 [205.0; 290.0]	0.082
CRP (mg/dL)	0.3 [0.1; 0.9]	0.2 [0.1; 0.4]	0.4 [0.2; 2.2]	<0.001
BUN (mg/dL)	14.9 [11.0; 19.0]	13.0 [10.0; 16.6]	16.2 [12.4; 22.0]	<0.001
Cr (mg/dL)	0.8 [0.7; 1.0]	0.7 [0.6; 0.9]	0.9 [0.8; 1.2]	<0.001
CK (U/L)	166.0 [88.5; 1194.5]	114.0 [75.0; 228.0]	424.5 [120.0; 2833.0]	<0.001
Troponin-I (ng/mL)	0.0 [0.0; 0.5]	0.0 [0.0; 0.1]	0.3 [0.0; 1.9]	<0.001
BNP (pg/mL)	26.0 [10.0; 89.0]	17.0 [9.0; 39.5]	53.0 [16.0; 137.0]	<0.001
Serum osmolality (mOsm/kg)	298.0 [291.0; 308.0]	296.0 [289.0; 307.0]	298.0 [293.0; 308.0]	0.154
Mortality	2 (22.2%)	0 (0.0%)	2 (1.2%)	0.232

Categorical and continuous variables are represented by number (%) and median ± interquartile range, respectively. The two groups were compared using the Mann–Whitney U test for continuous variables, and the chi-square test for categorical variables. HTN, Hypertension; DM, Diabetes mellitus; CKD, Chronic kidney disease; CAD, Coronary artery disease; LOC, loss of consciousness; COHb, Carboxyhemoglobin; WBC, White blood cell; Hb, Hemoglobin; CRP, C-reactive protein; BUN, Blood urea nitrogen; SD, standard deviation; Cr, creatinine; CK, creatinine kinase; BNP, brain natriuretic peptide; *p*-value < 0.05 is significant.

**Table 3 jpm-13-00921-t003:** Multivariable logistic regression to predict DNS occurrence in patients with CO poisoning.

	Crude OR (95% CI)	* *p*-Value	Adjusted OR (95% CI)	* *p*-Value
Age ≥65	1.51 (0.93–2.38)	0.08	0.83 (0.47–1.43)	0.51
History of CAD	4.10 (1.30–12.49)	0.01	3.16 (0.80–12.22)	0.10
LOC	2.17 (1.47–3.27)	<0.001 *	0.97 (0.60–1.59)	0.92
Mentality	1.78 (1.49–2.13)	<0.001 *	1.27 (1.01–1.60)	0.04 *
Seizure	0.88 (0.26–2.36)	0.823	1.19 (0.30–3.67)	0.78
HBOT	1.80 (1.09–3.12)	0.027*	1.38 (0.80–2.52)	0.27
WBC >12,550/mm^3^	2.63 (1.88–3.69)	<0.001 *	0.92 (0.58–1.42)	0.70
Troponin I >0.056 ng/mL	5.69 (4.01–8.17)	<0.001 *	2.12 (1.31–3.47)	0.002*
CK >341.5 U/L	6.35 (4.46–9.07)	<0.001 *	2.80 (1.81–4.33)	<0.001*
BNP >34.5 pg/mL	4.67 (3.30–6.63)	<0.001 *	2.28 (1.47–3.53)	<0.001*
Lactate >1.25 mmol/L	2.38 (1.63–3.57)	<0.001 *	1.70 (1.09–2.70)	0.021*

* *p* < 0.05 is significant. OR, odds ratio; CI, confidence interval; CAD, coronary artery disease; HBOT, hyperbaric oxygen therapy; CK, creatine kinase; BNP, brain natriuretic peptide.

## Data Availability

The datasets generated during the current study are available from the corresponding author on reasonable request.

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
