# Peer review of "Effectiveness of Initial Troponin I and Brain Natriuretic Peptide Levels as Biomarkers for Predicting Delayed Neuropsychiatric Sequelae in Patients with CO Poisoning: A Retrospective Multicenter Observational Study"

_jpm, 2023, doi:10.3390/jpm13060921_

Round 1

Reviewer 1 Report

The study entitled “Effectiveness of initial troponin I and brain natriuretic peptide levels as biomarkers for predicting delayed neuropsychiatric sequelae in patients with CO poisoning: A retrospective multicenter observational study.” is very interesting. Authors have tried to discover biomarkers through detailed analysis using appropriate statistical tools however there are major concerns in the paper as listed below.

According to the Table 1. Baseline characteristics of the study population, the Age (years) 41.0 [31.0;54.0] Total (n=967); 40.0 [31.0;52.0] Non-DNS (n=796) and 46.0 [35.0;58.0] DNS (n= 171) clearly indicating there is a very significant (<0.001) influence between the Non-DNS  and DNS.  Authors should prepare the age matched Non-DNS  and DNS groups to eliminate the impact. Most of the  Baseline characteristics of the study population in Table 1 are influenced by the significant (<0.001) age difference. After considering the aged matched groups authors can revise the table 1 and related contents. Aged match groups are important for ensuring that the results of studies are accurate and reliable. By comparing the levels of troponin I in aged match groups, researchers can be confident that any differences in the levels of troponin I are due to the factors that they are interested in studying, and not to other factors.

Troponin I levels are typically very high after a heart attack. Hence authors shall consider the significant (0.019) influence  by CAD

Authors should provide more detailed about then nonparametric data handling

Is there any Outliers due to wrong entries, how do they processed further on the Outliers,

No details provided about the missing data points

Minor suggestions

The “Figure 2. Receiver operator characteristic curve for prediction DNS using troponin I levels.” And “Figure 3. Receiver operator characteristic curve for prediction DNS using BNP levels.” Are not readable.

Discussion and conclusion are not very well elaborated and they are not coinciding with the observations.  Discussion and conclusion should be revised based on the recommended statistical analysis.

The authors also did not discuss the implications of the results of the study. They did not mention how the results could be used to improve the diagnosis and treatment

The authors should revise the discussion and conclusion of the paper to address these issues. They should discuss the results of the statistical analysis in detail and explain how the results support the hypothesis that age, gender, and CAD are all significant predictors of troponin I levels. By revising the discussion and conclusion of the paper to address these issues, the authors could improve the quality of their paper and make it more useful to other researchers.

Moderate editing needed

Author Response

Specific comments:

“The study entitled “Effectiveness of initial troponin I and brain natriuretic peptide levels as biomarkers for predicting delayed neuropsychiatric sequelae in patients with CO poisoning: A retrospective multicenter observational study.” is very interesting. Authors have tried to discover biomarkers through detailed analysis using appropriate statistical tools however there are major concerns in the paper as listed below.”

Dear Reviewer #1,

Thank you for your detailed and valuable comments. We considered all of your comments, as shown below.

Best regards.

Reviewer 2 Report

The authors investigated whether cardiac markers can be used as biomarkers to predict Delayed neuropsychiatric sequelae (DNS) occurrence following acute Carbon monoxide (CO) poisoning. In the present study, the authors found that higher Troponin I and brain natriuretic peptide (BNP) levels are associated with high-risk DNS patients. It is well written, presented, and discussed appropriately. However, there is room for improvement.

1)      Please provide directions for the result (higher levels of biomarkers) in the abstract that would help readers understand the study’s findings.

2)      Please provide a full of each abbreviation at the first appearance in the text.

Abstract

 CK,

BNP

Introduction

MRI

NSE

Methods

HBO

3)      Delayed neuropsychiatric sequelae (DNS) or Delayed neuropsychiatric syndrome (DNS). Please be consistent with your terminology.

4)      234 patients were dropped because of missing data. What variables were missing? Have you used those variables in the analyses or not?

5)      Author didn’t mention what biological sample was used for this study and when and how the sample was collected.

6)      I am curious, were all the continuous variables non-normally distributed?

7)      How were variables included in the multivariable regression model?

8)      Why were the following cut-offs used in the regression models?

Age ≥ 65

WBC >12,550/mm3

Troponin I >0.056ng/mL

CK>341.5 U/L

BNP>34.5 pg/mL

Lactate>1.25 mmol/L

9)      Author didn’t discuss how mentality, creatine kinase, and Lactate was implicated in DNS development.

Author Response

Specific comments:

“The authors investigated whether cardiac markers can be used as biomarkers to predict Delayed neuropsychiatric sequelae (DNS) occurrence following acute Carbon monoxide (CO) poisoning. In the present study, the authors found that higher Troponin I and brain natriuretic peptide (BNP) levels are associated with high-risk DNS patients. It is well written, presented, and discussed appropriately. However, there is room for improvement.”

Dear Reviewer #2,

Thank you for your detailed and valuable comments. We considered all of your comments, as shown below.

Best regards.
